# How Does High-Performance Work System Prompt Job Crafting through Autonomous Motivation: The Moderating Role of Initiative Climate

**DOI:** 10.3390/ijerph18020384

**Published:** 2021-01-06

**Authors:** Yuan Li, Xiyuan Li, Yujing Liu

**Affiliations:** Economics and Management School, Wuhan University, Wuhan 430072, China; liyuan0301@whu.edu.cn (Y.L.); liuyujing@whu.edu.cn (Y.L.)

**Keywords:** high-performance work system, job crafting, autonomous motivation, initiative climate, well-being

## Abstract

By invoking self-determination theory, we proposed an integrated, multilevel model to investigate the impact of a high-performance work system (HPWS) on employees’ job crafting through autonomous motivation, along with the moderation effect of initiative climate. Adopting a three-wave, time-lagged research design, we collected data from 615 employees of 54 Chinese companies. The results of multilevel path analysis revealed that (1) HPWS is positively related to employees’ job crafting; (2) HPWS has a positive impact on employees’ autonomous motivation; (3) employees’ autonomous motivation positively affects their job crafting; (4) employees’ autonomous motivation mediates the positive relationship between HPWS and employees’ job crafting; (5) initiative climate moderates the relationship between employees’ autonomous motivation and job crafting; and (6) the indirect relationship between HPWS and job crafting through autonomous motivation is also moderated by initiative climate. The findings of this study provided several implications for job crafting research and for human resource management in organizations.

## 1. Introduction

Although jobs are originally designed by organizations based on their requirements, job holders can also actively redesign their job characteristics according to their knowledge, techniques, capabilities, and incentives [1,2]. The process of shaping, molding, and changing job characteristics to make tasks and interpersonal interactions better fit an employee’s needs, capabilities, and preferences has been defined as job crafting [1,3,4]. Previous literature has indicated that job crafting can provide benefits for both employee well-being and organizational well-being [5,6,7]. Empirical evidence suggested that job crafting is positively related to several indicators for occupational well-being, such as work engagement [8,9], job satisfaction [8,10,11], psychological and subjective well-being [12], positive affect [11], as well as reducing burnout [13,14], job boredom [15], and job strain [8]. Prior studies also supported that job crafting can benefit organizations because it has a positive effect on the intention to stay [16], organizational commitment [13,17], organizational change [18], and task and contextual performance [8,19]. Given that enacting job crafting is an effective approach to prompt both individual and organizational well-being as well as the further development, it has recently attracted considerable research interest [7].

However, while most of the previous literature concentrated on the consequences of job crafting, the examination of the antecedents of job crafting is still neglected [4,20]. Few prior studies explored the effect of individual differences (e.g., big five personality [8]; proactive personality [21]; approach/avoidance temperament [22,23]), job characteristics (e.g., job autonomy [8,24]; task interdependence [4,25]; playful work design [26]), motivation characteristics (e.g., self-efficacy [8]; work engagement [27]), and leadership styles (e.g., transformational leadership [28,29]; empowering leadership [30,31]; servant leadership [32]) on job crafting, but the research that investigated the effect of the organizational environment, such as the high-performance work system (HPWS), on job crafting is still scarce. Correspondingly, job crafting theory has underlined the important role of the work environment in facilitating employee’s job crafting [1,33]. Due to the nested nature of the work environment, employees’ attitudes and behavior will be inevitably affected by others and the context in which they are embedded. Thus, it contributes to expanding our understanding of job crafting through incorporating organizational environment factors into job crafting research with multilevel perspective.

Precisely, in this research, we intend to illustrate the relationship between the organizational work environment, focusing on the implement of HPWS and employees’ tendency to engage in job crafting, as well the potential mechanism and boundary conditions. As a set of human resource management practices aiming to foster workforce ability, motivation, and opportunity, HPWS has been demonstrated to have positive effects on broadly-defined proactive behavior [34]. However, there is no literature that shed light on the relationship between HPWS and job crafting, though the latter has been known as a specific and valuable type of initiative behavior. Thus, to address this important research gap, the present study mainly focused on three research questions: (1) Does HPWS have positive influence on employees’ job crafting? (2) What is the underlying mechanism between HPWS and employees’ job crafting? (3) Which situation can prompt the effect of HPWS on employees’ job crafting?

Specifically, the first target of this research is examining the influence of HPWS on employees’ job crafting. Moreover, according to self-determination theory, employees will generate more autonomous motivation when their basic psychological needs are satisfied, which in turn facilitates their initiative and engagement [35,36,37]. Meanwhile, the practices of HPWS can fulfill employees’ needs for competence, autonomy, and relatedness, thereby fostering their autonomous motivation. Therefore, given that autonomous motivation may transmit the positive impact of HPWS to employees’ job crafting, the second aim of this research is investigating the mediating effect of autonomous motivation between the high-performance work system and job crafting. Furthermore, based on person-situation theory, the surrounding environment provides reinforcement or barriers when employees are motivated to achieve various goals [38]. We supposed that the initiative climate, which is a specific work setting to support personal initiative, may play a facilitating role in the process of the above-mentioned mediating relationship. Thus, the third goal of this research is to explore the moderating role of the initiative climate on the mediation effect of employees’ autonomous motivation between the HPWS and job crafting.

Our study provides three theoretical contributions. Firstly, this research advances the literature on job crafting by investigating whether HPWS is a facilitating organizational environment for employees’ job crafting. Second, by adopting self-determination theory, we investigated the mediation role of autonomous motivation on the relationship between the HPWS and job crafting, thereby promoting our understanding regarding the mediating mechanism about how HPWS influences job crafting. Finally, we contribute to the job crafting literature by investigating the boundary conditions for the forming of job crafting, and found that the initiative climate can moderate the indirect effect of the HPWS on employees’ job crafting through autonomous motivation.

## 2. Theoretical Background and Hypotheses

### 2.1. High-Performance Work System and Job Crafting

High-performance work system (HPWS) refers to a set of human resource practices that are interconnected internally and consistent with the organizational strategy externally, aiming to promote workforce ability, motivation, and opportunity, and further improving organizational performance. [34,39,40]. Specifically, a high-performance work system includes three dimensions: (1) Skill-enhancing practices, such as selective staffing and skill training; (2) motivation-enhancing practices, which include long-term, results-oriented appraisal and extensive, open-ended rewards; and (3) empowerment-enhancing opportunities, such as encouragement of participation. By utilizing these integrated HR practices, employees obtain KSAs (knowledge, skills, and abilities), motivation and opportunities to carry out specific beneficial behavior, which provides a long-term competitive advantage for the organization [41]. In this research, we focus on the relationship between HPWS and employees’ job crafting, and suppose that a HPWS is positively associated with employees’ job crafting for several reasons.

Firstly, through various training and promoting practices, HPWS are implemented to enhance employees’ knowledge, skills and abilities [41,42]. As job crafting means an improvement of work content, it is necessary for employees to acquire relevant KSAs to trigger job crafting. For example, Cullinane et al. [43] demonstrated that skill utilization is a crucial facilitator for daily job crafting. Therefore, by developing employees’ KSAs, HWPS can increase the likelihood of their participant in job crafting. Secondly, motivation-enhancing practices, such as results-oriented appraisal and incentive reward system, encourage employees to work hard for higher reward, which leads to promoting their motivation [44]. The high level of motivation enables employees to proactively perform their work with novel approaches, which will give rise to the likelihood of change in their work resources and task boundaries [37]. Previous literature has illustrated the positive correlation between motivation-enhancing practices and job crafting. For instance, Meijerink et al. [45] indicated that motivation-enhancing HR practices facilitate employee to proactively increase their challenging job demands, which is an important form of job crafting.

Finally, the empowerment-enhancing practices of HPWS, such as flexible job assignments and encouragement of participation, can foster employees’ perception of autonomy in their work [24,34]. When employees gain the autonomy to select the specific ways in which to implement their tasks and obtain a perception of empowerment, they tend to craft the job boundaries and overall attitude of their work [24]. Recent studies have provided an empirical evident relationship between job autonomy and job crafting [7,8,44]. Thus, motivation-enhancing initiatives of HWPS might motivate employees to participate in job crafting by increase their autonomy. Taken together, the practices of HPWS may have a joint effect on employees’ job crafting.

**Hypothesis 1 (H1).** 
*HPWS is positively related to employees’ job crafting.*


### 2.2. The Mediating Role of Autonomous Motivation

As the above discussion, we argue that HPWS is a positive predictor for employees’ job crafting by linking HPWS with the job crafting literature. Moreover, there is an important underlying motivation mechanism that we intend to emphasize is the employees’ autonomous motivation, which refer to “behaving with a sense of volition and possessing the experience of choice” [46]. Based on self-determination theory (SDT), all individuals have three fundamental psychological needs, which are the need for competence, autonomy, and relatedness [46]. SDT defines the need for competence as the desire to explore and control the environment, as well as to seek the best challenges; the need for autonomy represents individuals’ demand to have a sense of ownership of their behavior and feel mentally free; and, finally, the need for relatedness refers to the desire for developing an intimate relationship with some others, as well as feeling loved and cared for by others [47]. Gagne and Deci indicated that the work environment that facilitate satisfaction of these innate needs will drive employees’ autonomous motivation [46]. In other words, if individuals believe that their psychological needs are satisfied, it can be concluded that they are autonomously motivated. According to SDT, we suppose HPWS would foster the employees’ autonomous motivation by satisfying their needs for competence, autonomy, and relatedness.

HPWS includes practices such as extensive training and development in the organization, which can stimulate employees’ autonomous motivation by fulfilling the employees’ need for competence. As discussed above, training and promoting practices of HPWS are effective to improve employees’ knowledge, skills, and abilities [41,42], while such enhancements contribute to strengthening individuals’ perceptions of work-related competence [48]. Moreover, HPWS can also promote the employees’ autonomous motivation by satisfying their needs for autonomy. The promotions, results-oriented appraisal, open-ended rewards, and flexible job assignments of HPWS can enhance employees’ perceptions of autonomy and influence at work [49]. Employee participation of HPWS can foster a sense of choice and volition by supplying employees with opportunities to participate in the work arrangement [39], which contributes to enhancing employees’ perception of control at work.

Finally, the encouragement of participation within HPWS can strengthen employees’ feeling of communion and belonging by encouraging them to work together with colleagues. The collaborative work environment provides employees’ opportunity to build close personal relationships with other people through frequent cooperation and communication at work. When individuals consider themselves as a member of the team, and establish harmonious relationships with other team members, they are likely to feel satisfied with their need for relatedness [47]. To sum up, we conclude that HPWS will have a positive effect on employees’ autonomous motivation, as we propose the following hypothesis:

**Hypothesis 2 (H2).** 
*HPWS is positively related to employees’ autonomous motivation.*


As noted above, job crafting is a self-initiated behavior that employees take to shape, mold, and change some aspects of their job [7]. Individuals, therefore, need a high level of motivation to embark on this course. Due to autonomously motivated employees having more experience of meaningfulness, choice, competence, and are more likely to insist on new and challenging complex tasks [46], as well as to concentrate on those tasks [50], we argue that they might be more possible to engage in job crafting.

Specifically, autonomously motivated employees reveal themselves more easily to increase structural job resources and social job resources because a high level of autonomous motivation helps them hold sufficient interest and enthusiasm to make an effort for the improvement of work [5,46]. For instance, Lee and Song [37] demonstrated that intrinsic motivation, which is one type of autonomous motivation, has a positive effect on developing employees’ passion and positive feelings for their work, helping them to expand task and relational boundaries creatively. Moreover, according to self-determination theory, autonomously motivated employees are concerned with their growth and learning at work, and they are more likely to seek higher levels of challenge for self-enhancement. Autonomous motivation also facilitates high levels of vitality and proactivity for employees when they deal with tasks [50], which contribute to setting more challenging goals and persist in achieving these difficult goals.

Finally, autonomously motivated employees may implement decreasing hindering job demands as an approach to deal with work that is of no value or meaning to them. Minimizing unnecessary or hindering demands help individuals concentrate on the tasks that provide more benefits for their growth and learning [29]. There is empirical evidence to prove that reducing hindering job demands have a positive effect on personal growth and meaningful performance for individuals [51]. In conclusion, we suppose that employees’ autonomous motivation may have a positive effect on employees’ job crafting:

**Hypothesis 3 (H3).** 
*Employees’ autonomous motivation is positively related to their job crafting.*


Taken together, HPWS may positively affect employees’ autonomous motivation, which, in turn, may positively affect job crafting by employees. Therefore, we propose:

**Hypothesis 4 (H4).** 
*Employees’ autonomous motivation mediates the positive relationship between HPWS and employees’ job crafting.*


### 2.3. The Moderating Role of Initiative Climate

Since employees carry out their work in an organizational context, they will be influenced by environmental information during the process from psychological cognition to behavior generation [52]. Organizational climate is an important situational factor for employees’ attitude and behavior, because it determines the shared perceptions of what manners and demands are appropriate, and which behavioral patterns that are expected and motivated in the environment [53]. Therefore, when employees believe self-initiated job design is expected and encouraged in their organization, they are more able to craft some aspects of their job. Among multiple types of organizational climate, such as the service climate [54], justice climate [55], and safety climate [56], we focus on initiative climate, which is a specific work setting to support personal initiative, and investigate its facilitative effect on the forming of employees’ job crafting.

Initiative climate refers to “employee shared perceptions of the extent to which self-starting, change-oriented, long-term oriented, and persistent behavior is encouraged and rewarded by management” [53] (p. 653), which has been proved as an effective work environment to foster individual proactive behavior [57,58]. The basic principle of person–situation theory indicates that the surrounding environment provides reinforcement or barriers when employees are motivated to achieve various goals [38]. When there is a high level of initiative climate in the organization, employees will generate a perception that they are expected and encouraged to enact self-initiated, change-oriented, future-oriented behavior. The situational cue provided by the initiative climate are consistent with employees’ autonomous behavioral tendencies, thereby reinforcing these autonomous motivated employees’ tendencies to shape, mold, and change some aspects of their job.

On the contrary, when the level of initiative climate is low in the organization, the environmental cue reveals that self-initiated behavior is undesired or unaccepted, therefore restricting the enacting of job crafting by employees. Consequently, the variance between contextual signals and behavioral tendency of autonomously motivated employees weaken the effect of employees’ autonomous motivation on their job crafting. In view of the above discussion, we suppose that the initiative climate moderates the relationship between autonomous motivation and job crafting in such a way that the relationship is stronger at a higher level of initiative climate. Thus, we put forward the following hypothesis:

**Hypothesis 5 (H5).** 
*Initiative climate moderates the relationship between employees’ autonomous motivation and job crafting, such that the positive relationship is stronger for high levels of initiative climate than low levels of initiative climate.*


Furthermore, we propose a moderated mediation model (as shown in Figure 1), that is, the indirect effect of HPWS on employees’ job crafting through employees’ autonomous motivation will be moderated by the initiative climate. General speaking, when the level of initiative climate is higher, the positive relationship between employees’ autonomous motivation and job crafting is stronger; thus, the more influence of HPWS will be conveyed to employees’ job crafting through autonomous motivation. On the other hand, when the level of the initiative climate is lower, the positive relationship between employees’ autonomous motivation and job crafting is weaker; therefore, the impact of HPWS on job crafting will be less conveyed through employees’ autonomous motivation.

**Hypothesis 6 (H6).** 
*Initiative climate moderates the indirect effect of HPWS on job crafting through employees’ autonomous motivation, such that the indirect effect is stronger for high levels of initiative climate than low levels of initiative climate.*


The conceptual model of this research is depicted in Figure 1.

## 3. Methods

### 3.1. Participants and Procedure

Our final usable sample consisted of 615 full-time employees from 54 firms across three provinces of China. The firms are mostly from traditional manufacturing, financial, transportation, and high-tech industries. The selected firms all have a series of clear management systems and employees have certain decision-making power over their own work, that is, there is the possibility and opportunity of job crafting. In terms of firm size, they range from a hundred employees to two thousand employees. With approval from the top management of these firms, the human resource (HR) managers helped us invite employees to participate. We sent messages to HR managers with website links containing questionnaires, and asked them to finish the HR manager questionnaire and distribute the employee questionnaire. Our study complied with research and publication ethical guidelines, and we explained the purpose of our study to all respondents and assured them the anonymity and confidentiality. To improve the response rate, we offered a reward of RMB30 (approximately US $5) to each respondent who completed the entirety of the questionnaire.

Data were collected at three time points and from two different data resources. At Time 1, we sent a message to HR managers with website links containing measures of HPWS. At Time 2, one month later, we sent messages to HR managers with website links containing the HR manager questionnaire and the employee questionnaire. The HR manager questionnaire contained measures of organizational-level initiative climate and the employee questionnaire contained measures of individual-level autonomous motivation, proactive personality, and basic demographic information. At Time 3, also one month later, we sent a website link containing measures of individual-level job crafting to all employees who submitted at Time 2.

At Time 1, a total of 56 HR managers responded to questionnaires (93.3% response rate). At Time 2, we distributed a survey to 689 employees from 54 HR managers, leading to response rates of 96.4% and 91.9%, respectively. At Time 3, we received 615 employee questionnaires (89.3% response rate). Therefore, we obtained completed matched data from 54 HR managers and 615 employees. The industry type statistics of 54 firms is shown in Table 1. A total of 35.1% of the 615 employees was female and 64.9% was male. The majority had obtained at least a college degree (76%). The average age of the participants was 32.65 years (SD = 6.92), and the average organizational tenure was 4.32 years (SD = 4.85).

### 3.2. Organizational-Level Measures

We used the translation-back-translation procedure to translate original English scales into Chinese [59]. All survey items were measured on a five-point Likert scale rated from 1 (“strongly disagree”) to 5 (“strongly agree”).

*High-performance work system*. HR managers in each establishment evaluated respective organizational HPWS with the 27-item measure developed by Sun et al. [39]. Sample items include “In our company, long-term employee potential is emphasized” and “In our company, employees are provided the opportunity to suggest improvements in the way things are done” (α = 0.871).

*Initiative climate*. We assessed organizational initiative climate via HR manager ratings with the seven-item measure developed by Baer and Frese [60]. Sample items were “In our company, employees actively attack problems” and “In our company, employees usually do more than they are asked to do” (α = 0.712).

### 3.3. Individual-Level Measures

*Autonomous motivation*. We assessed employees’ autonomous motivation with the 6-item measure developed by Gagné et al. [61]. Participants indicated to what extent did they feel following motivations when chosen the current job. Sample items were “I chose this job because it fits my personal values” and “I chose this job because because I enjoy it very much” (α = 0.956).

*Job crafting*. We assessed employees’ job crafting with the 21-item measure developed by Tims et al. [3]. Sample items were “I make sure that my job is mentally less intense” and “I ask others for feedback on my job performance” (α = 0.953).

*Control variables*. We controlled for individual demographics (i.e., employees’ age, gender, education level, and organizational tenure). Considering active dispositional variables may affect job crafting [21,58], we also controlled for proactive personality. Therefore, we assessed employees’ proactive personality with the four-item measure developed by Parker et al. [62]. An example item was “I am excellent at identifying opportunities” (α = 0.822).

### 3.4. Analytic Approach

We conducted multilevel path analysis with Mplus 7.4 [63], because the mediation and moderation hypotheses in our study are all crossing organizational and individual levels. Specifically, to test Hypothesis 4, we calculated the indirect effects with the parametric bootstrapping method and obtained confidence intervals based on Monte Carlo simulations with 20,000 replications using the open-source software R 4.0.0. Such method is followed the suggestion by Preacher et al. [64] and preferred over the normal distribution-based significance tests. In testing the cross-level moderation in Hypothesis 5, we readjusted employees’ autonomous motivation with group-mean centering and organizational-level initiative climate using grand-mean centering [58].

## 4. Results

### 4.1. Preliminary Analysis

To provide support for the discriminant validity of variables in our study, we conducted a confirmatory factor analysis (CFA) to examine the distinctiveness among individual proactive personality, autonomous motivation, and job crafting which self-rated by employees. The hypothesized three-factor model fit the data acceptably (χ^2^ = 771.048, *df* = 74, RMSEA = 0.124, CFI = 0.910, SRMR = 0.049) and was better fit than one-factor model with all individual-level variables loaded on a single factor (χ^2^ = 1746.164, *df* = 77, RMSEA = 0.188, CFI = 0.785, SRMR = 0.086). Although the RMSEA value of three-factor model was slightly above the commonly acceptable discrimination criteria [65], the model satisfied the criteria of CFI and SRMR values still exhibiting an acceptable fit to the data [58]. Table 2 shows the descriptive statistics and correlations among study variables. Notably, HPWS were significantly correlated with individual job crafting (r = 0.183, *p* < 0.001) and autonomous motivation (r = 0.233, *p* < 0.001). Individual autonomous motivation also significantly correlated with their job crafting (r = 0.756, *p* < 0.001).

### 4.2. Hypotheses Testing

Table 3 provides the estimated results from the multilevel path analysis. In support of Hypothesis 1 and 2, we found that organizational-level HPWS had a significant positive cross-level influence on individual job crafting (*γ* = 0.266, *p* < 0.01) and autonomous motivation (*γ* = 0.597, *p* < 0.001), respectively. This result (as shown in Model 2 and Model 1) indicates that employees’ job crafting and autonomous motivation were significantly higher in organizations with higher HPWS. Individual autonomous motivation was positively related to job crafting (*γ* = 0.356, *p* < 0.001) as shown in Model 3, supporting Hypothesis 3. We examined the mediating roles of autonomous motivation for the cross-level relationship between organizational-level HPWS and individual-level job crafting. We used the parametric bootstrap method based on 20,000 Monte Carlo replications to test for the indirect effect and we found that HPWS was associated with employees’ job crafting through autonomous motivation (indirect effect = 0.262, 95% bias-corrected bootstrap confidence interval [0.119, 0.407]). Since the CI did not include zero, Hypothesis 4 was supported.

Moreover, the result of Model 4 in Table 2 depicts that the cross-level interaction between organizational initiative climate and employees’ autonomous motivation had a positive and statistically relationship with individual job crafting (*γ* = 0.243, *p* < 0.001). To further test the cross-level moderation effect, we plotted the interactive effects at high (i.e., +1 SD) and low (i.e., −1 SD) levels of initiative climate, as shown in Figure 2 [66]. We also conducted a simple slope analysis following the recommendations by Preacher et al. [67]. The results showed that the positive relationship between autonomous and job crafting was stronger for those who had high levels of organizational initiative climate (*γ* = 0.448, SE = 0.033, *p* < 0.001) and weaker for those who had low levels of initiative climate (*γ* = 0.258, SE = 0.019, *p* < 0.001). Hence, Hypothesis 5 was supported.

Last, we examined the indirect effects of employees’ autonomous motivation on the relationship between HPWS and job crafting as moderated by organizational-level initiative climate. We found that the mediating relationship between HPWS and job crafting via autonomous motivation was stronger under a high level of initiative climate (0.231, *p* < 0.001) but were weaker under a low level of initiative climate (0.135, *p* < 0.001). The difference in the indirect effect between two conditions was 0.096 (95% CI [0.033, 0.159]), supporting Hypothesis 6.

## 5. Discussion

In summary, the present study proposed a multilevel model investigate the potential relationship between HPWS and job crafting. We found that there is significant positive effect of HPWS on employees’ job crafting. Moreover, our findings demonstrated that the influence of HPWS on job crafting is mediated by employee autonomous motivation, supporting the hypothesis we proposed in this study. In addition, the result also illustrated that the indirect influence of HPWS on employees’ job crafting is moderated by initiative climate, more specifically, the indirect relationship between HPWS and job crafting is stronger when there is a high level of initiative climate. There are several implications for theoretical research and managerial practice that we concluded below.

### 5.1. Theoretical Contributions

Regarding to theoretical contributions, first of all, the present study advanced the literature on job crafting by investigating whether HPWS is a significant predictor for employees’ job crafting. Although job crafting literature has emphasized the important role of work environment in motivating job crafting by employees [1], we still know little about the influence of macro work environment on job crafting [37]. At the same time, HRM practice literature indicated that high-performance work system can foster employees’ broadly defined proactive behavior through facilitating work workforce ability, motivation and opportunity [34]. By linking prior job crafting literature and HRM practice literature, we supposed there is an underlying correlated relationship between HPWS and job crafting that has not been tested in previous research. Therefore, in this research, we focused on an important organizational work context, which is HPWS, and explored its effect on employees’ job crafting. The findings of this study supported the hypothesis that organizational-level HPWS had a salient positive cross-level effect on individual job crafting. It can be demonstrated that a bundle of high-performance HRM practices can significantly enhance employees’ tendency to engage in job crafting, which contribute to extend the research of the contextual antecedents of job crafting. Meanwhile, the present study also provided insight for the organizational level facilitators of employees’ job crafting by cross-level analysis, responding to current calls for job crafting study to consider contextual influences and to use multilevel perspectives [8,14].

Second, by adopting self-determination theory, we investigated the mediation effect of autonomous motivation on the relationship between HPWS and job crafting, thereby enhancing our understanding about the underlying mechanism through which HPWS influence job crafting. In this study, we integrated self-determination theory into the job crafting literature, and proposed that autonomous motivation is a key mechanism for the linking of high-performance HRM practices and change-oriented behaviors. Our findings supported the mediating role of autonomous motivation for the cross-level relationship between organizational-level HPWS and individual-level job crafting, providing empirical evident for the principles of self-determination theory, which suggested that employees will obtain more autonomous motivation when their basic psychological demands are fulfilled, which, in turn, they are more likely to enact self-initiative and change-oriented behavior [36,37]. These results illustrated that when the work environment satisfies employees’ need for competence, autonomy, and relatedness, employees will believe their behavior is self-determined and engage in behavior to craft job resources and task boundaries. The findings of this research provided empirical evidence and verified researchers’ viewpoints that motivation has significant influence on facilitating employees’ job crafting [1,68].

Finally, we broadened the job crafting literature by exploring the boundary conditions for the forming of job crafting, and found that the initiative climate can moderate the indirect effect of HPWS on employees’ job crafting through autonomous motivation. The forming of job crafting is a contextually embedded phenomenon, therefore, it is vital for job crafting research to illustrate the boundary conditions of job crafting [7]. By adopting person-situation theory [38], we proposed that initiative climate is a facilitating factor for both the direct influence of autonomous motivation and the indirect effect of HWPS on employees’ job crafting. The results supported our proposed our hypothesizes and indicated that that the indirect influence of HPWS on job crafting through autonomous motivation is stronger under the level of initiative climate is higher, but is weaker under the level of initiative climate is lower. This finding confirmed the moderating effect of initiative climate in the forming process of job crafting. This is also in line with the person-situation theory, which suggests that an encouraging work environment can provide reinforcement for employees to achieve improvement [38]. Our findings provide empirical evidence that the contextual factor can moderate the influence of multilevel antecedents on job crafting, responding to recent calls for the further investigating boundary conditions of job crafting [7,8,14].

### 5.2. Practical Implications

This research also provides some practical implications for organization management. First, because employees’ job crafting is an important driver of organizational well-being and employees’ well-being, organizations should take actions to foster employees’ job crafting. The findings of this research support that HPWS has a positive influence on job crafting by employees and, therefore, indicate that the implementing of high-performance HRM practices is an effective approach to foster employees’ job crafting. Skill-enhancing practices, motivation-enhancing practices, and empowerment-enhancing practices should be implemented by managers in organizations, such that they provide multiple resources and chances for employees to craft their job resources and task boundaries. Managers should pay close attend to the integration of each practice in HPWS, because the synergetic implementation of individual HRM practices will make the full use of HPWS.

Second, enterprises should pay attention to employees’ autonomous motivation, because autonomous motivation is not only a key antecedent of employees’ job crafting, but also an important mediation mechanism for high-performance HRM practice to improve job crafting by employees. In order to facilitate employees’ job crafting effectively, companies can improve employees’ autonomous motivation through implementing a bundle of high-performance HRM practices aiming to fulfill employees’ need for competence, autonomy, and relatedness. Moreover, managers could also satisfy employees’ psychological need by making reasonable task arrangement, offering discretion for subordinates, and creating a collaborative atmosphere.

Last, we also find that initiative climate is a crucial boundary condition for the indirect influence of HPWS on job crafting through autonomous motivations in present study. Thus, we advise that managers ought to encourage employees to engage in self-initiated, future-oriented, and endurable behaviors, and formulate incentive plans to motivate employees to participant in these behaviors. Meanwhile, managers should be fully aware of every employees’ individual characteristics and work demands by in-depth communication, and provide various kind of encouragement or assistant for targeted employees. By doing that, employees will feel that self-initiated job design is expected and encouraged in their organization and are more possible to enact job crafting which can provide benefits for both organizational and individual well-being.

### 5.3. Limitations and Future Research Directions

In addition to these theoretical contributions and practical recommendations, the present research also has a few limitations that need to be demonstrated. Firstly, we use an approach of “HR manager-evaluation” to measure the high-performance work system and initiative climate. However, this approach only reveals the HR managers’ subjective assessment for HPWS and initiative climate, therefore, the authenticity and accuracy of this research may be interfered. Future research can adopt more effective measurement methods to assess high-performance work systems and initiative climate, such as collecting both of supervisor-reported data and employee-reported data. Secondly, given that we only collect data in a Chinese context, the generalizability of the present study to other cultural contexts should be concerned. Future research can replicate our study across a wider range of cultural settings to examine the extent to which the present study can be applied in other culture contexts. Thirdly, we only investigate the underlying mechanism of autonomous motivation for the impact of HPWS on employees’ job crafting. However, there might be other mediation mechanisms, such as role breadth self-efficacy and work passion, to link the relationship between HPWS and employees’ job crafting. Hence, we hope future research can provide further insight for underlying mechanisms and boundary conditions of the relationship between HPWS and employees’ job crafting by adopting other theories. Fourthly, the potential influence of the industrial context on the effect of initiative climate should be taken into consideration in future research. Since we mainly focused on the moderating role of initiative climate in the proposed theoretical model, we did not investigate the detailed information about the industrial environment in which these firms are embedded. Future research would benefit by further examining the effect of initiative climate in different industrial context.

Finally, our study mainly concerned the overall construct of job crafting, and did not examine the influence of HPWS and autonomous motivation on different subdimensions of job crafting. In recent years, based on regulatory focus theory and approach-avoidance motivation theory, job crafting scholars suggested job crafting can be both enrich and enlarge, or lessen and limit, which illustrated that both approach crafting and avoidance crafting are subdimensions of job crafting [7,23,69]. Approach crafting is an initiative behavior to seek for positive components of a job, whereas avoidance crafting can be regarded as an adaptive behavior to avoid negative components of a job. We suggest future research should concern the difference between subdimensions of job crafting. Meanwhile, because approach crafting differs from avoidance job crafting in aspects of behavioral characteristics, there might be discrepancies among their generation mechanisms and contextual factors. For instance, employees may tend to engage in approach crafting in an autonomous environment, while they are likely to enact avoidance crafting in controlled environments. Future research can investigate which contexts are more effective to prompt approach crafting or avoidance crafting, to further enhance our understanding about the boundary conditions of different types of job crafting.

## 6. Conclusions

Drawing upon self-determination theory, the present study found that HPWS was a significant organizational-level predictor for employees’ job crafting. In addition, the result demonstrated that employees’ autonomous motivation was a key underlying mechanism to link the relationship between HPWS and job crafting. Moreover, the findings also illustrated that initiative climate was a crucial boundary condition for the indirect influence of HPWS on job crafting through employees’ autonomous motivation.

## Figures and Tables

**Figure 1 ijerph-18-00384-f001:**
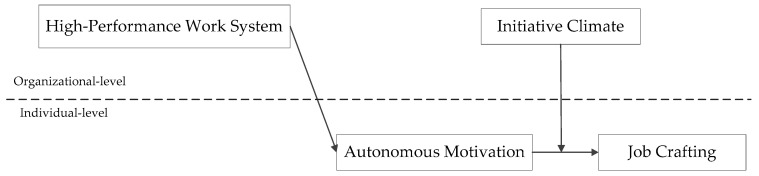
Research model.

**Figure 2 ijerph-18-00384-f002:**
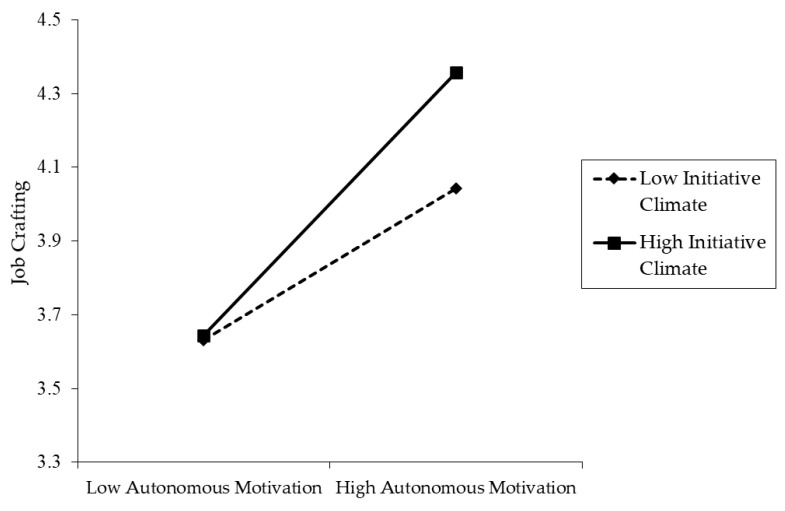
Interactive effects of individual-level autonomous motivation × organization-level initiative climate on job crafting.

**Table 1 ijerph-18-00384-t001:** Sample industry type statistics.

Industry	Number
Traditional manufacturing	18
Financial	12
Transportation	9
High-tech	7
Consulting	3
Real estate	3
Pharmaceutical	2

Note: N = 54.

**Table 2 ijerph-18-00384-t002:** Descriptive statistics and correlations among study variables.

Variables	Mean	SD	1	2	3	4	5	6	7	8	9
*Individual-level*											
1. Age	1.350	0.478	—								
2. Gender	32.650	6.921	−0.100 *	—							
3. Education level	2.900	0.772	0.092 *	−0.205 ***	—						
4. Organizational tenure	4.323	4.857	0.022	0.487 ***	−0.401 ***	—					
5. Proactive personality	3.754	0.629	−0.141 ***	−0.082 *	−0.028	−0.048	—				
6. Autonomous motivation	3.784	0.798	−0.130 **	−0.066	0.037	0.000	0.584 ***	—			
7. Job crafting	3.955	0.543	−0.071	−0.112 **	0.000	−0.050	0.711 ***	0.756 ***	—		
*Organizational-level*											
8. HPWS	3.882	0.314	−0.069	0.151 ***	−0.021	0.084 *	0.100 *	0.233 ***	0.183 ***	—	
9. Initiative climate	3.551	0.389	0.003	0.238 ***	−0.313 ***	0.339 ***	−0.005	0.034	0.090 *	0.488 ***	—

Note: For individual-level, n = 615; for organizational-level, N = 54. HPWS = high-performance work system. * *p* < 0.05. ** *p* < 0.01. *** *p* < 0.001.

**Table 3 ijerph-18-00384-t003:** Multilevel path analysis results.

Variables	Autonomous Motivation		Job Crafting					
	Model 1		Model 2		Model 3		Model 4	
	Estimate	SE	Estimate	SE	Estimate	SE	Estimate	SE
Intercepts	3.675 ***	0.053	3.917 ***	0.037	3.955 ***	0.012	3.919 ***	0.037
*Organizational-level*								
HPWS	0.597 ***	0.138	0.266 **	0.095				
Initiative climate							0.210 **	0.072
*Individual-level*								
Age	0.001	0.056	0.035	0.034	0.063 *	0.026	0.021	0.028
Gender	−0.009	0.005	−0.005 **	0.002	−0.002	0.002	−0.003	0.002
Education level	0.000	0.045	0.001	0.021	−0.022	0.018	0.003	0.013
Organizational tenure	0.027 ***	0.007	0.007	0.004	−0.003	0.003	−0.003	0.002
Proactive personality	0.652 ***	0.045	0.565 ***	0.036	0.352 ***	0.024	0.323 ***	0.024
Autonomous motivation					0.356 ***	0.019	0.353 ***	0.021
*Cross-level interaction*								
Initiative climate × Autonomous motivation							0.243 ***	0.042

Note: For individual-level, *n* = 615; for organizational-level, N = 54. SE = standard error. HPWS = high-performance work system. * *p* < 0.05. ** *p* < 0.01. *** *p* < 0.001.

## Data Availability

Data available on request due to privacy.

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
