# Peer review of "How Does High-Performance Work System Prompt Job Crafting through Autonomous Motivation: The Moderating Role of Initiative Climate"

_ijerph, 2021, doi:10.3390/ijerph18020384_

Round 1

Reviewer 1 Report

Thank you for the opportunity to review this article. The time involved in submitting your manuscript is greatly appreciated.

Despite this, the article presents a series of issues that must be noted and mended. The recommendations are presented separately by sections. Hopefully, they would be useful.

Title: the title does not adequately reflect the content of the paper. Please, try to change it to better inform the readers about that relationships between the variables that you test and also inform about the quality of your sample. It is a good point for you because it is large, so it seems better to inform the readers about it in the title.  

Abstract:

Less information appears in the abstract. Maybe expanded by adding the most relevant findings. Please, take into account that the abstract is the unique part of your paper that most of the readers could read. Hence, more information would be better. 

Introduction:

Firstly, some of the references that you cite are too old (in example, 1985, 1989) . Even though the most relevant studies should be referenced, also the RECENT research must be included.

At the end of the literature review, the aims and the questions in the research should appear. Maybe to formulate the questions as a hypothesis would be an option to clear this aspect. Another commentary, it is the possibility of including this part at the final of the introduction part; even a separate section could be a good option, in order to clear the final of the introduction and to serve as a connection with the method.

Method:

Please, try to better describe the sociodemographic data of your participants. In the same sense, give the readers detailed information about the procedure for recruiting participants and collecting data.

Related to the instruments, please better inform about their psychometric quality and give to the readers some examples of the items. If you can, please inform me about previous studies where the same instrument has been used and the reliability obtained in that research.

Data analyses

Please, explain to the readers which procedures of statistical analyses have been used and justify your decisions.

Results

The results should be presented in the same order as the introduction and hypotheses. Also, the same order must be used in the Tables. This simplifies the work for readers.

Finally, the repetition is constant all over the article. Please, try to change the words in order to do the reading more interesting and motivating

Discussion:

First of all, try to better adjust your conclusions to the findings. Or to say in other words, please try to justify more clearly the connection between your conclusions and your findings.

Finally, a section related to limitations, future lines of investigations and the principal contributions of the research could be interesting. Your paper has a lot of relevant implications for society and policymakers, but you need to elaborate more on this topic.

Conclusion:

They don’t appear new conclusions on this part. This part does not add any new to the rest of the paper. Please, try to condense your findings, or to highlight your main contribution to the field.

Author Response

Thanks for the reviewer’s comment. The authors have revised the manuscript according to the reviewers' comments. Please see the attachment.

Reviewer 2 Report

The paper under review examines an interesting question, that of how high-performance work systems prompt job crafting, focusing on the moderating Role of initiative climate, based on data from employees from 54 Chinese firms. Overall the paper is well-written; the analysis seems to be well-executed. However, there are some puzzles lingering in the paper.

First, more information on the sampling strategy should be provided: Which three provinces were selected? Why these three? How were the 54 firms selected? By what criteria? What industries do they represent (-- the paper briefly mentions that the firms come from such and such industries, but why not make a table to detail the profile of the study sample)? What are the sizes of the firms? Also unclear is the type of employees surveyed: Are they entry-level staff, or they are managers? Without these pieces of information, it's hard to assess the externality of your findings. 

Second, I am not so sure whether self-determination theory, a theory that was developed while observing firms in western countries, is suitable for analyzing Asian firms, which are often subject to governmental control. 

Third, it's hard to understand the magnitude of the estimated effects (which is a general problem with structural equation modeling and path analysis, etc.)

Fourth, does initiative climate play different roles in different industries? I would envision that it plays a bigger role for firms that are more competitive and have more potential for promotion, etc. 

Fifth, the "policy" recommendations provided by the paper seem not specific enough. Yes, you found a high level of X leads to a higher value of Y, so you recommend that one should try to raise the level of X. But how (and is it feasible)? More specific recommendations may need to be provided to make these recommendations useful in practice. 

Author Response

(The authors gave the same response as above.)

Reviewer 3 Report

The paper deals with an important topic, one of current research interest, and it does offer new knowledge in the field of the discussed phenomena. The Authors are testing the relationships between High-Performance Work System and job crafting through autonomous motivation. Initiative climate was assumed as a moderator of variables. Although the reviewed article is cognitively interesting and generally well-written, it contains several weak points which should be eliminated in the further stages of working on its text. Thus, in the following part of the review I focus more on the weaker points of the article than on its merits, although the former do not undermine the undoubtedly valuable and methodologically well-prepared article. Some remarks are to be found below, and these have general nature.

General remarks:

  • Although, as the Authors state, it is true that a significant number of researchers examining the issues of crafting have focused on the consequences of job crafting, it is necessary to point out that a number of research projects have also been conducted with regard to its antecedents. This is evident, for instance, in the meta-analyses by Rudolph et al. (2017), cited by the Authors, as well as in the studies of Zhang et al. (2019), or Lee, Lee (2018). However, I agree with the Authors that the influence of HRM on job crafting has not yet been explained adequately. From the point of view of the theoretical framework, the variables described in the article need some supplementation. Although the Authors sufficiently justified the selection of HPWS with regard to AMO (ability, motivation, opportunities), the conceptualization of the variable of job crafting still requires a broader discussion. We learn from the section on Methods that this variable has been assessed with the use of the 21-item measure developed by Tims at al. (2012), based on Job Demands-Resources Theory (Bakker, Demerouti, 2007). In the light of the concepts of Tims, Bakker, Derks (2012), the increase of the person-job fit forms the base for undertaking job crafting. Motivational aspects connected with realization of needs are emphasized in the approach of Wrzesniewski, Dutton (2001), who distinguish cognitive crafting, relational crafting and task crafting. Why, then, do the Authors refer to the former approach and how do they justify such a choice?
  • Another issue is connected with the model and the selection of constructs. The Authors indicate autonomous motivation as the main mechanism of undertaking job crafting, while ignoring the situational factors which may lead to crafting, in particular with regard to avoidance-oriented job crafting, or in a situation when the change in tasks forces employees to gain resources (precisely within the scope of job crafting) in order to increase task efficiency or job fit. In such cases job crafting has an adaptive character and, therefore, is not pro-active, which was mentioned by Bruning and Campion (2018). In certain situations, adapting to job conditions and achieving task performance may require the employee to decrease hindering job demands. I think that it would be worthwhile to mention this issue in the paper, as the Authors diagnosed job crafting with the use of four dimensions: increasing structural job resources, increasing social job resources, increasing challenging job demands, decreasing hindering job demands.
  • Initiative Climate – adopted as a moderator of the relationship in the research model, it is not an exogenous variable because we are speaking here about how the climate is perceived. On the other hand, in the light of the definitions and the theoretical framework of job crafting adopted by the Authors, it is necessary to remember that it also covers behaviours which consist in the avoidance of demands which are limiting in nature (avoidance crafting). Therefore, the problem of a lack of cohesion may arise. When speaking about initiative and proactivity, it is necessary to keep in mind, preferably, the other three dimensions: increasing structural job resources, increasing social job resources, increasing challenging job demands.
  • The methodology is generally appropriate. I have only a few questions to this section:

What was the method for selecting companies for the study? What was the structure of the studied companies (e.g. size, industry, etc.)?

In general, the methods used and explained are appropriate for the design of the study. The results of the studies were presented and analysed sufficiently. The verification of the research hypotheses was also conducted correctly. I perceive the aggregation of the literature as especially successful.

Author Response

(The authors gave the same response as above.)

Round 2

Reviewer 2 Report

I have no further comments on the substance of the paper, but please check the Enligh carefully -- I found language problems at times.